# Augmenting zero-Kelvin quantum mechanics with machine learning for the prediction of chemical reactions at high temperatures

Jose Antonio Garrido Torres[1,2], Vahe Gharakhanyan [2,3], Nongnuch Artrith [1,4,5], Tobias Hoffmann Eegholm[1] & Alexander Urban [1,2,5✉]

The prediction of temperature effects from first principles is computationally demanding and typically too approximate for the engineering of high-temperature processes. Here, we introduce a hybrid approach combining zero-Kelvin first-principles calculations with a Gaussian process regression model trained on temperature-dependent reaction free energies. We apply this physics-based machine-learning model to the prediction of metal oxide reduction temperatures in high-temperature smelting processes that are commonly used for the extraction of metals from their ores and from electronics waste and have a significant impact on the global energy economy and greenhouse gas emissions. The hybrid model predicts accurate reduction temperatures of unseen oxides, is computationally efficient, and surpasses in accuracy computationally much more demanding first-principles simulations that explicitly include temperature effects. The approach provides a general paradigm for capturing the temperature dependence of reaction free energies and derived thermodynamic properties when limited experimental reference data is available.

[1] Department of Chemical Engineering, Columbia University, New York, NY 10027, USA. [2] Columbia Electrochemical Energy Center, Columbia University, New York, NY 10027, USA. [3] Department of Applied Physics and Applied Mathematics, Columbia University, New York, NY 10027, USA. [4] Columbia Center for Computational Electrochemistry, Columbia University, New York, NY 10027, USA. [5] Materials Chemistry and Catalysis, Debye Institute for Nanomaterials Science, Utrecht University, 3584 CG Utrecht, The Netherlands. ✉email: a.urban@columbia.edu

The decarbonization of chemical industry is a necessity for the transition to a sustainable future[1–3], but developing alternatives for established industrial processes is cost intensive and time consuming. Bottom-up computational process design from first-principles theory, i.e., without requiring initial input from experiment, would be an attractive alternative but has so far not been realized. On the other hand, computational materials design and discovery based on atomic-scale first-principles calculations has already become commonplace and is a powerful complement to experimental materials engineering[4,5]. Here, we demonstrate how first-principles quantum-mechanics based theory can be supplemented with a machine-learning (ML) model describing temperature dependence to enable the prediction of chemical reactions at high temperatures.

Temperature effects are especially important for chemical and electrochemical reactions that involve reactants and products in different states of matter, such as corrosion reactions (i.e., the binding of oxygen in a solid oxide)[6,7] or the reverse, the extraction of metals from their oxides. As one example, we focus here on the latter and consider the pyrometallurgical reduction of metal oxides. In industry, many base metals, such as cobalt, copper, and silver, are extracted from their ores via smelting, using carbon as the reducing agent[8,9]. Recycling of transition and rare earth metals, e.g., from spent batteries and electronics waste, also commonly involves pyrometallurgical processes[10,11]. However, our findings apply more generally also to other classes of reactions at high temperatures.

Given their relevance, an inexpensive computational method for predicting the temperature dependence of oxidation or oxide reduction reactions would be extremely attractive. Empirical models based on the parametrization of experimental thermodynamics data, such as the Calculation of Phase Diagrams (CALPHAD) approach[12,13], have been used for the thermodynamic characterization of materials at different temperatures and for virtual process optimization[14] but are limited by the amount of available data from experiments. First-principles (quantum-mechanics based) calculations provide efficient and reliable estimates of ground-state materials properties at zero Kelvin[15,16]. Introducing temperature effects increases the computational cost of the simulations by several orders of magnitude, which is not amenable for the screening of large numbers of compositions and thermodynamic conditions required to aid with process optimization[17]. Hence, there is a need for computational methods that exhibit the computational efficiency of an interpolation-based method such as CALPHAD and the transferability of first-principles methods. We will demonstrate in the following that ML techniques can provide the missing link.

A growing body of literature evidences that first-principles modeling can be greatly accelerated by training ML models on the outcome of first-principles calculations[18–20]. However, in many cases, accurate data for high-temperature materials properties cannot be readily generated from first-principles calculations, and experimental thermochemical databases are much smaller in size. For example, we were only able to compile a set of 38 metal oxide reduction temperatures from public data sources that were extracted from experimentally measured free energies of reaction (see Supplementary Table 1). In the case of such data limitations, it is crucial for the construction of accurate models to make use of prior knowledge, for example, in the form of known laws of physics or thermodynamics.

In the present work, we show that combining both information from first-principles calculations and data from experiment can enable the construction of quantitative models for the prediction of temperature-dependent materials properties such as metal-oxide reduction temperatures (Fig. 1a). The key novelty of our approach is that it makes use of known thermodynamic relationships (Fig. 1b). The predictions from an ML model based on Gaussian process regression (GPR)[21] and results from first-principles calculations both enter the thermodynamic equations that govern metal oxide reduction, enabling the quantitative prediction of high-temperature materials properties of oxides that were not included in the reference data set (Fig. 1d). Through this thermodynamic underpinning, other temperature-dependent physical properties can be accessed at no extra cost and with a higher accuracy than when training ML models directly for specific observables. In particular, we demonstrate that the zero Kelvin first-principles calculations can be augmented with machine-learned temperature effects to yield a physics-based ML model for predicting high-temperature reaction free-energies with great accuracy but at a low computational cost.

As a specific example, we consider the pyrometallurgical reduction of metal oxides to their base metals, though our general approach can be expected to be transferable also to other high-temperature reactions. Specifically, we aim at predicting the reduction temperature of metal oxides ($M_xO_y$) using carbon coke as a reducing agent, which corresponds to the chemical reaction

$$M_xO_y(s) + yC(s) \rightleftharpoons xM(l, s) + yCO(g), \qquad (1)$$

where M is the base metal (or a mixture of multiple metal species) of a given metal oxide $M_xO_y$, CO is carbon monoxide, and $x$ and $y$ are the corresponding stoichiometric reaction coefficients. In general, the metal oxide and carbon are in their solid state, while the reduced metal can be liquid (smelting) or solid and carbon monoxide is in the gas phase.

The Gibbs free energy of reaction corresponding to Eq. (1) can be expressed as

$$\Delta_r G_{red}(M_xO_y) = x\Delta_f G(M) + y\Delta_f G(CO) - \Delta_f G(M_xO_y) - y\Delta_f G(C), \qquad (2)$$

where $\Delta_f G(M)$, $\Delta_f G(CO)$, $\Delta_f G(M_xO_y)$, and $\Delta_f G(C)$ denote the Gibbs free energy of formation of M, CO, $M_xO_y$, and C, respectively. At room temperature, most metal oxides are highly stable and the equilibrium of reaction (1) is strongly tilted to the left-hand side, i.e., $\Delta_r G_{red}(M_xO_y)$ is positive. But at high-enough temperatures, the greater entropy of gas phases compared to solids shifts the equilibrium to the right-hand side of Eq. (1), making the reduction energetically favorable, i.e., $\Delta_r G_{red}(M_xO_y)$ becomes negative.

Our objective is the computational prediction of the reduction temperature $T_{red}$ above which the sign of $\Delta_r G_{red}(M_xO_y)$ becomes negative and reduction of the metal oxide occurs.

In the following, we will compare three different computational approaches: (i) a fully non-empirical approximation of $T_{red}$ based only on first-principles density-functional theory (DFT); (ii) a ML model obtained from a direct fit of experimental reduction temperatures; and (iii) a hybrid scheme that augments DFT zero-Kelvin predictions with an ML model of the temperature-dependent contributions.

A series of approximations is required to arrive at a purely first-principles estimate of the reduction temperature. The temperature dependence of the Gibbs free energy of formation of an oxide compound X, $\Delta_f G(X) = \Delta_f H(X) - TS$, is partly due to the temperature dependence of the enthalpy of formation $\Delta_f H$ but mostly stems from the entropy term $TS$, where $T$ is the temperature and $S$ is the overall entropy. At zero Kelvin, the entropy term vanishes and the Gibbs free energy of formation is identical to the enthalpy of formation, which can be directly obtained from DFT calculations. For example, the zero-Kelvin formation enthalpy of the metal oxide can be calculated as

$$\Delta_f H_{M_xO_y}^{DFT}(T = 0\ K) = E_{M_xO_y}^{DFT} - xE_M^{DFT} - \frac{y}{2}E_{O_2}^{DFT}, \qquad (3)$$

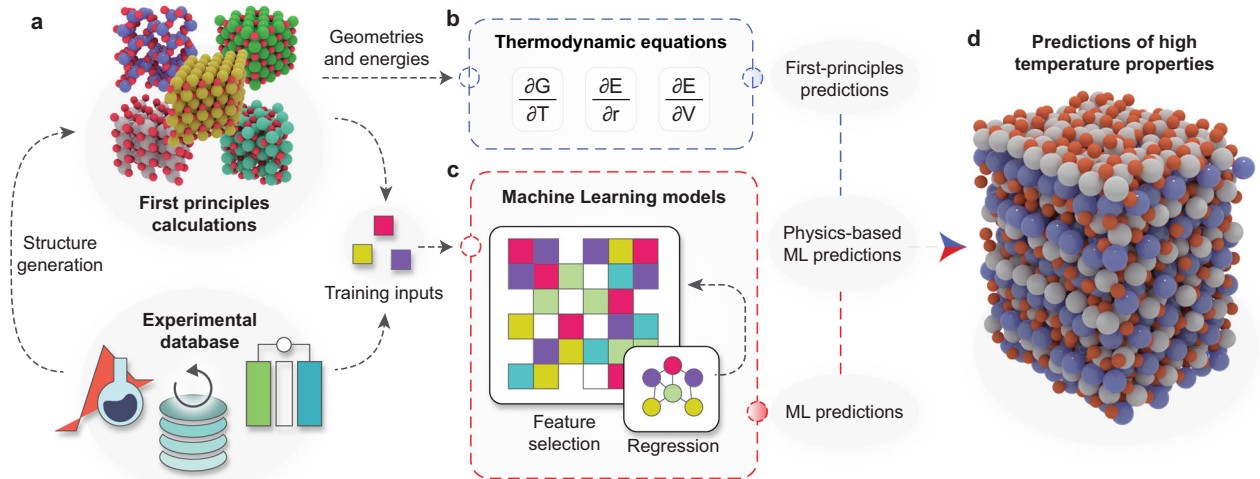

**Fig. 1 Hybrid model for predicting high-temperature properties of metal oxides. a** Zero Kelvin first-principles calculations and finite-temperature experimental data of metal oxides from the literature are compiled in a database. **b** Thermodynamic quantities are evaluated from first principles where feasible. **c** Features are extracted from the data set and used as inputs to build a quantitative machine learning (ML) model for those contributions that are not accessible from first principles. **d** The output from the ML model and from first-principles calculations together enter a physics-based framework for the prediction of the temperature-dependent stability of other metal oxides that were not included in the reference data set.

where $E_{M_xO_y}^{DFT}$, $E_M^{DFT}$, and $E_{O_2}^{DFT}$ are the DFT energies of the metal oxide, the base metal, and an oxygen molecule in the gas phase, respectively. The greatest contribution to the temperature-dependent terms of the reaction free-energy (2) arises from the entropy of the molecular gas species CO ($S_{CO}$), which can be efficiently approximated in the ideal gas limit from first-principles calculations. The vibrational entropy of the solid phases at a given temperature can also be obtained from first principles by integration of the vibrational density of states, which can be approximately obtained from DFT phonon calculations[22]. Additional contributions to the free-energy arise from the electronic, magnetic, and configurational entropies[23,24], which can also be approximated from first principles but have not been considered in the present work. Further details of the DFT calculations and additional entropy contributions are given in the "Methods" section.

The experimental reduction temperature values along with the corresponding predictions obtained from DFT calculations are shown in Fig. 2. DFT reduction temperatures are shown for an approximation only accounting for the entropy of CO and including additionally the vibrational entropy contributions from phonon theory. See Supplementary Table 2 and Supplementary Fig. 1 for the corresponding data and correlation plots of the predicted and reference reduction temperatures. As expected, the accuracy of the reduction temperatures increases when a higher level of theory is included in our model: the mean absolute error (MAE) and the root mean squared error (RMSE) of the DFT-based models decrease when including phonon corrections to the free-energy from 235 K to 166 K and from 265 K to 202 K, respectively. However, including phonon corrections is computationally demanding and scales poorly with increasing number of atoms, making it computationally expensive for crystal structures with large unit cells. Due to this high computational cost, we computed 38 samples (binary and ternary oxides) using DFT but limited phonon calculations to the 19 binary oxides only.

As seen in Fig. 2, even on the highest level of theory considered, the predicted first-principles reduction temperatures are on average still subject to large errors of around 200 K. Including phonon calculations improved the DFT predictions across the board, but the relative error reduction is barely significant except

for the compounds MgO, CaO, SiO$_2$, and TiO$_2$. Considering the high computational cost of phonon calculations, this result is sobering and reflects both the approximations made in the form of the reaction free energy and the intrinsic error of DFT.

The limited accuracy of the first-principles models motivated us to explore whether ML models can predict oxide reduction temperatures with superior accuracy. Intuitively, the temperature-dependent vibrational entropy contributions are determined by the nature of the chemical bonds in the various oxides, i.e., we expect differences depending on the degree of ionic and covalent character[25]. As input for the ML model, we therefore chose features that affect the chemical bonding and can be easily obtained from the periodic table or by means of efficient DFT calculations. The following properties were used for the construction of compound fingerprints:

(i) Atomic properties: atomic number ($Z$), atomic mass ($m$), electronegativity ($\chi$);
(ii) Bond properties: ionic character ($I_C$);
(iii) Composition properties: oxidation state (Ox), stoichiometry ($\phi$);
(iv) Structure properties: unit cell volume ($V$), density ($\rho$), center of mass ($\mu$); and
(v) Phase properties (from DFT): 0 K formation enthalpy ($\Delta H_f$), bulk modulus ($B_0$).

Note that only two of the properties are derived from DFT calculations, the formation energies and bulk moduli, the calculations of which are straightforward and computationally efficient. The construction of the compound fingerprint by combining the above properties is described in the Supplementary Methods section and in Supplementary Table 3. We employed recursive feature elimination to detect redundant features and avoid overfitting as is detailed in the Supplementary Methods section and Supplementary Fig. 2.

We trained a GPR-based ML model on the experimental reduction temperatures of Supplementary Table 1 and quantified its accuracy using leave-one-out cross-validation (LOOCV). LOOCV ensures that the model is evaluated only for samples that were not used for training and is a standard technique for assessing the transferability of a model (see further details in the

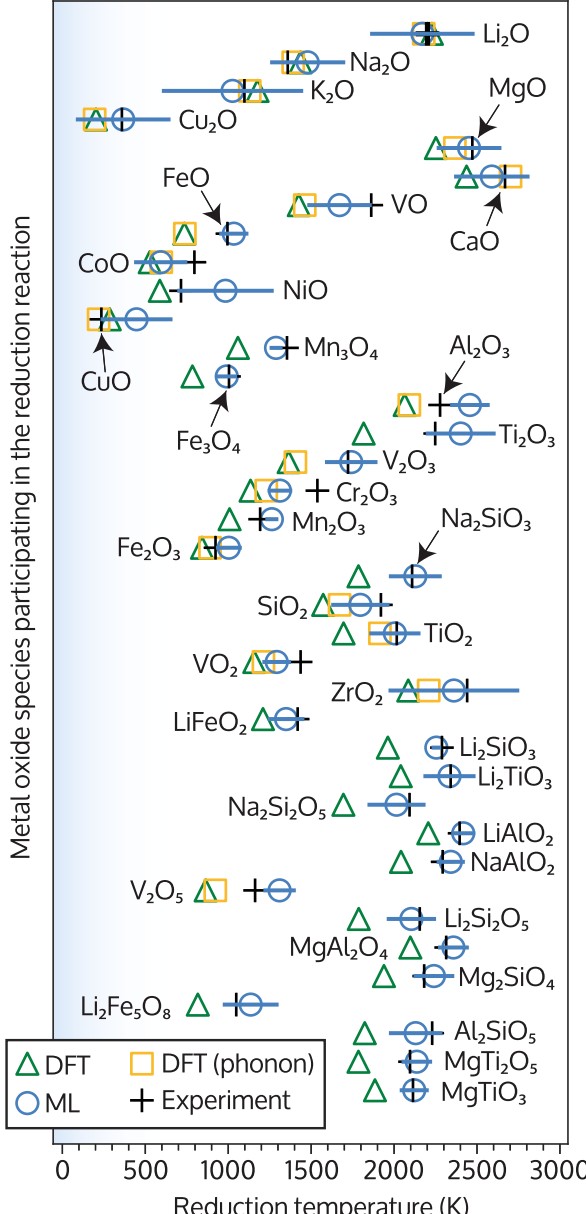

**Fig. 2 Comparison of predicted and reference metal oxide reduction temperatures.** Green triangles indicate the oxide reduction temperatures as predicted by density-functional theory (DFT) calculations only considering the temperature-dependent entropy contributions of CO. The orange squares show the reduction temperatures after correcting the DFT calculations using phonon theory. The predictions obtained from our machine learning (ML) model trained on reduction temperatures are indicated by blue circles. These data points were obtained from leave-one-out cross-validation and are thus pure predictions and were not included in the model construction. The horizontal error bars indicate the uncertainty estimates from the Gaussian process regression model. The black crosses indicate the experimental reference values of Supplementary Table 1. See Supplementary Fig. 1 for correlation plots of the predicted temperatures with the reference values. All data can be found in Supplementary Table 2.

"Methods" section). Further details of the model hyperparameters and construction are given in the methods section. In addition, we also performed multiple rounds of cross-validation using different partitions to study the robustness of the predictive power with respect to the train/test fold size (see Supplementary

Fig. 3). The predicted reduction temperatures from LOOCV for each sample are compared with their corresponding experimental reference in Fig. 2.

We observe that the predicted reduction temperatures of the GPR model surpass in accuracy the first-principles values obtained when using only DFT, even when computationally expensive phonon corrections were included. The MAE and RMSE from LOOCV are 105 K and 127 K, respectively, which is around 50% smaller than the errors of the pure DFT predictions. In addition to a greatly improved predictive power, another benefit of the Gaussian process regression model compared to DFT is the uncertainty estimate that it provides (shown as blue error bars in Fig. 2).

In the first instance we used only experimental data from binary metal oxides (19 samples) to train and validate the ML model to allow for a fair comparison with the DFT results including phonon theory. Since the first-principles calculations needed for building the compound fingerprint are computationally affordable, we expanded our reference data set from 19 to 38 samples by including ternary oxides as well. LOOCV on the larger data set shows that including extra reference data improves the predictive power of the GPR model further and reduces the MAE and RMSE by 20 K and 18 K to 85 K and 109 K, respectively. In contrast, the DFT-based model (without phonon correction) showed larger errors for the increased data set (MAE and RMSE increased by 21 K and 12 K, respectively), presumably because of the neglected entropy contributions that become more relevant with increasing number of constituents.

Metal extraction via carbothermal reduction becomes technically challenging if the oxide reduction temperature is significantly above 1500 K, in which case other processes such as hydrometallurgical routes are more commonly used[26]. Focusing on pyrometallurgy, a useful model would have the ability to predict reduction temperatures below 1500 K with an accuracy of at least around ±100 K. Unfortunately, the direct ML model predictions are the least accurate for the relevant temperature range. Additionally, there is no guarantee that the direct ML model correctly captures the underlying thermodynamic principles that govern metal reduction, since we have treated the GPR model akin to a black box. To further validate the model in this respect, we considered the competing chemical reactions that are at the core of pyrometallurgical processes.

The overall reaction of Eq. (1) can be understood as a competition between the formation reactions of carbon monoxide and the metal oxide

$$C + 1/2 O_2 \rightarrow CO \text{ and } xM + \frac{y}{2} O_2 \rightarrow M_x O_y, \qquad (4)$$

where metal oxide formation is energetically more favorable at low temperatures, and CO formation is favored at high temperatures. The metal oxide reduction temperature is then determined by the intercept of the free energies of the CO and metal oxide formation reactions when normalized to the same oxygen content, i.e., the temperature for which $2 \Delta_f G(CO) = \frac{2}{y} \Delta_f G(M_x O_y)$. This relationship of CO and oxide formation energies is visualized in Ellingham diagrams[27], which are a common tool for the engineering of pyrometallurgical processes.

Figure 3a–c show Ellingham diagrams for a subset of the considered compounds as predicted by DFT (with and without phonon corrections) and the direct ML model compared to experimentally measured references from the NIST-JANAF[28] and Cambridge DoITPoMs[29,30] databases. Free-energy diagrams for the entire set of compounds are shown in Supplementary Fig. 4.

Ellingham diagrams contain more information than the reduction temperature, as they represent the variation of reaction free energies (per $O_2$ molecule) across different temperatures:

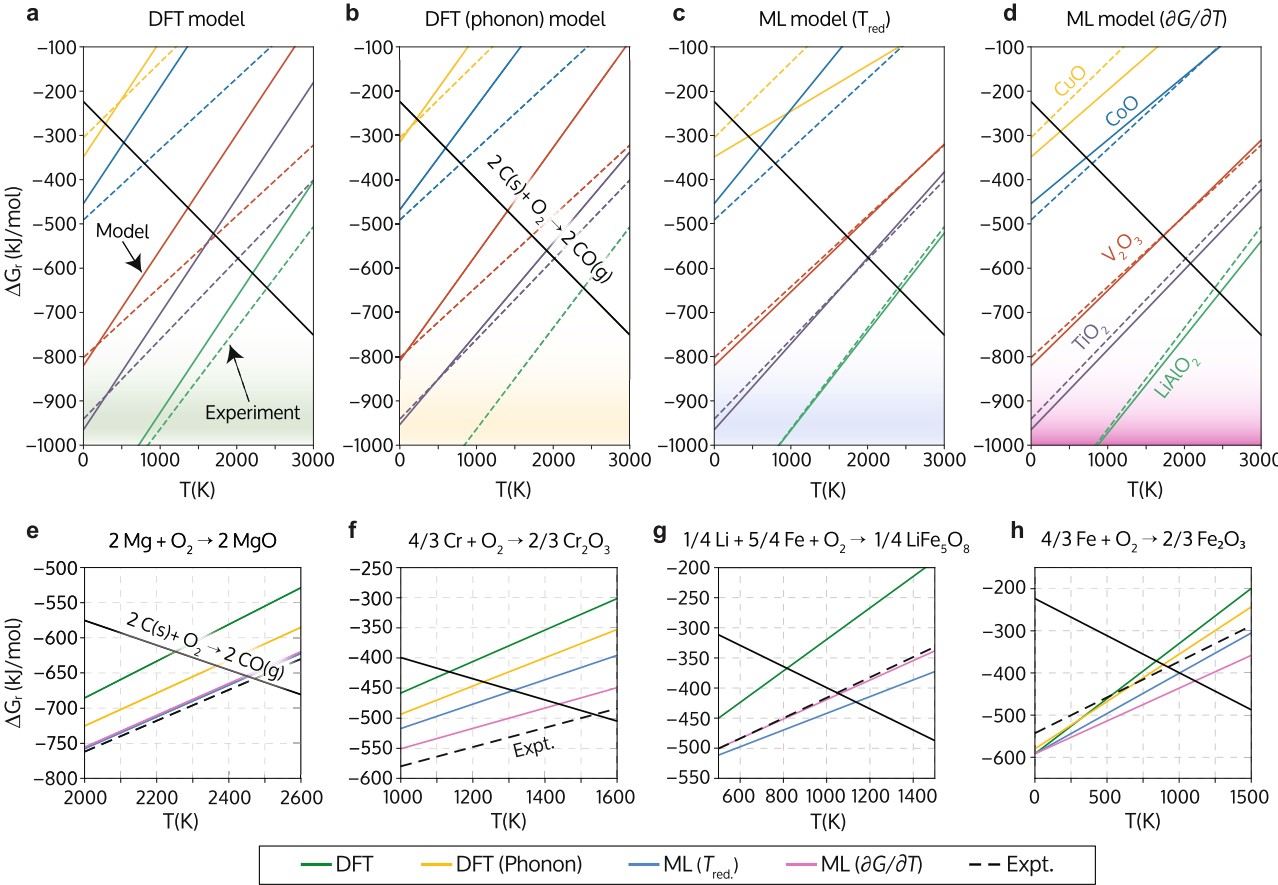

**Fig. 3 Predicted reaction free-energy curves for different metal oxides and models. a–d** Predicted Ellingham diagrams obtained from **a** density-functional theory (DFT), **b** DFT including phonon theory, **c** the machine-learning (ML) model trained on reduction temperatures, and **d** the hybrid ML model trained on metal oxide formation free-energy slopes ($\partial G/\partial T$). The labels for the different metal oxides are included in panel **d**. Predicted free-energy curves are represented by the solid colored lines whilst the dashed lines are the experimental reference values (Expt.). The free-energy curves for the formation of **e** MgO, **f** $Cr_2O_3$, **g** $LiFe_5O_8$, and **h** $Fe_2O_3$ obtained from DFT (green), DFT using phonon corrections (orange) and the ML models trained on reduction temperatures (blue) and free-energy slopes (pink) are compared with their corresponding experimental reference data (black dashed lines). The black solid lines indicate the experimental free-energy curve for CO formation.

(i) Reduction temperatures can be determined from the intersection of the free energy of formation of CO (black line with negative slope) and the formation energy of a given metal oxide (positive slope), (ii) the reaction enthalpies at zero Kelvin ($\Delta_f H(T=0K)$) are given by the $y$-intercepts of the oxide formation free energies, and (iii) the change of the formation free-energies with respect to the temperature ($\partial \Delta_f G/\partial T$, in the following $\partial G/\partial T$ for conciseness) is the slope of a metal oxide formation free-energy curve in the Ellingham diagram.

As seen in Fig. 3a, b, the slope of the reaction free energy curves predicted purely from first principles deviates significantly from the experimental reference, indicating that the temperature-dependence of the oxides is subject to large errors. The direct ML model trained on the reduction temperatures $T_{red}$ (Fig. 3c) is in excellent agreement with experiment for $V_2O_3$, $TiO_2$, and $LiAlO_2$, both in terms of the slope of the reaction free energies as well as for the intercept with CO formation. However, the ML model predicts a completely wrong temperature dependence for CuO and CoO.

The failure of the $T_{red}$ ML model for some of the compounds is due to errors in the DFT zero-Kelvin formation enthalpies of CuO and CoO. Even though the ML model predicts reduction temperatures for the two oxides that are close to the reference, the temperature dependence is described incorrectly because of large errors in the $y$-intercept of the reaction free energy curves,

i.e., the zero-Kelvin enthalpies. The $T_{red}$ ML model thus does not capture the underlying thermodynamics correctly and would not predict useful free energies at temperatures other than the oxide reduction temperature. Hence, the $T_{red}$ ML model does not provide reliable predictions of the temperature-dependent reduction free energy and would therefore not be useful for, e.g., the prediction of reduction potentials for high-temperature electrolysis[31].

To further improve the predictive capabilities of our model we decomposed it into two parts: zero Kelvin formation energies that can be obtained from DFT according to Eq. (3) and the temperature variation of the oxide formation free energy ($\partial G/\partial T$). Training targets for this hybrid ML model are thus the experimental values for $\partial \Delta_f G/\partial T$. Figure 3d shows the Ellingham diagram obtained from the combination of the DFT zero Kelvin formation energies and the ML-predicted free energy change with the temperature. This hybrid ML model predicts the temperature dependence of the formation free energies of CuO and CoO in excellent agreement with experiment and simultaneously improves the accuracy of the reduction temperature predictions. The remaining error is mostly due to the zero-Kelvin formation energy and no longer is an artifact of the ML model.

As previously for the $T_{red}$ model, we validate the $\partial G/\partial T$ model by comparing the experimental reduction temperatures with the values obtained by our predictions using LOOCV. Training on

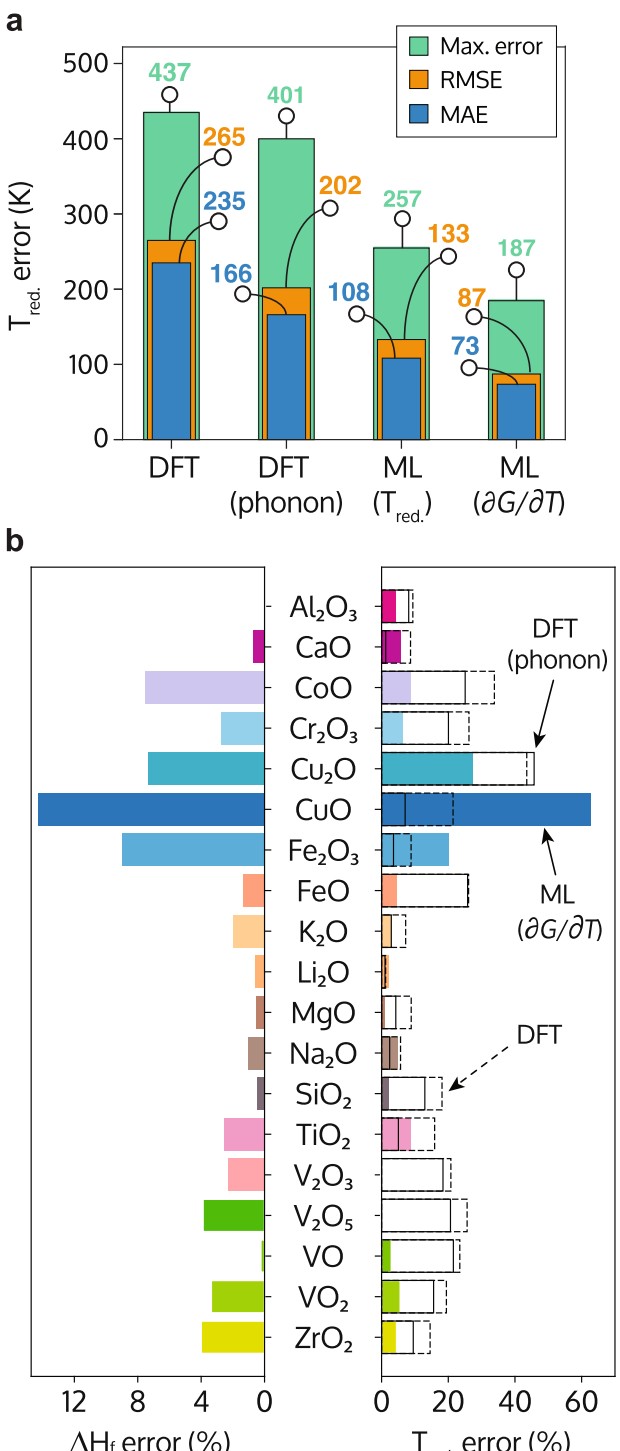

**Fig. 4 Performance metrics for the DFT and ML theoretical models.**
**a** Mean absolute error (MAE, blue bars), root mean square error (RMSE, orange bars), and maximum error (green bars) of the theoretical models. **b** Comparison between the formation energy ($\Delta H_f$) and reduction temperature ($T_{red}$) errors committed by the theoretical methods: density-functional theory (DFT, empty bars), DFT including phonon corrections (empty dashed bars) and the physics-based machine learning (ML) model trained for $\partial G/\partial T$ (solid color bars).

the experimental $\partial G/\partial T$ values of the binary oxides decreases the cross-validation scores even further, yielding an MAE of 74 K and an RMSE of 87 K, which are 31 K (MAE) and 40 K (RMSE) smaller than the errors of the $T_{red}$ model (Fig. 4a). The LOOCV

scores of the larger data set including ternary oxides are yet smaller, giving an MAE and RMSE of 64 K and 78 K, respectively, showing that the predictive power of the $\partial G/\partial T$ model increases as samples are added to the data set.

Finally, we analyze the accuracy of the predicted free-energy curves for different metal oxides to determine the source of the errors of the various models. In the interest of space, we only include four representative examples here and refer the reader to Supplementary Fig. 4 for the free-energy curves of all 38 samples considered in this work. This holds for the majority of the considered oxides, especially for the ML model trained for $\partial G/\partial T$, which produces the most accurate predictions for the free-energy curves. We note that the ML methods can be applied to large structures, e.g., ternary oxides (Fig. 3g), for which the phonon corrections are computationally too demanding.

One of the advantages of training the ML model on the free energy slopes $\partial G/\partial T$ is that it is possible to distinguish between errors in the zero Kelvin formation enthalpy and errors in the predicted temperature dependence committed by the ML model. For some metal oxides, such as $Fe_2O_3$, the zero Kelvin enthalpy of formation is not well described by DFT (Fig. 3h). In this example, the reduction temperatures obtained from DFT and from the $T_{red}$ model are close to the experimental reference, but only because of a compensation of errors in the zero-Kelvin enthalpy and the free energy slope. The $\partial G/\partial T$ model predicts the temperature dependence of the free energy in much better agreement with experiment than the other models, and the remaining error in the reduction temperature of $Fe_2O_3$ is dominated by the DFT error in the zero-Kelvin enthalpy. Hence, the $\partial G/\partial T$ model correctly captures the underlying physics.

In fact, as seen in Fig. 4b, the errors in the DFT zero-Kelvin formation energies correlate with the errors in the reduction temperatures predicted by the $\partial G/\partial T$ model. This does not hold true for the other models. The $\partial G/\partial T$ model decouples the zero Kelvin energies from the DFT calculations and the ML predictions of the temperature-dependent terms. As a consequence, increasing the accuracy of the zero-Kelvin enthalpies, e.g., by means of a higher level of theory, would yield an immediate improvement of the model accuracy.

This is further corroborated by a baseline ML model combining the experimental zero-Kelvin formation enthalpies with the ML-predicted free-energy slopes. This baseline model, which by construction does not exhibit the DFT errors at zero Kelvin, yields indeed an improved accuracy reducing the MAE and RMSE of the oxide reduction temperatures from 64 K and 78 K to 52 K and 65 K, respectively. An overview of the LOOCV error estimates of all discussed models is compiled in Supplementary Table 4. Since the feature vector of the ML model contains information derived from DFT, we would expect even further improvements in the predictions when using more accurate first-principles methods. Together, this test indicates that the accuracy of the hybrid ML model can still substantially improve if a more accurate electronic-structure method becomes available.

A limitation of our model arises from the fact that phase transitions were only implicitly included in the reference data, i.e., the model is not aware of the melting points of the base metals even though some of the metals melt well below the reduction temperature of their oxides. This is not a deficiency of the model for the prediction of reduction free energies and reduction temperatures. However, we believe that models for other temperature-dependent processes can be built following the same principles that we put forward here but training with explicit phase information, i.e., by distinguishing between different solid, liquid, and gas phases. We further note that an extension of our model to compositions with greater number of species is straightforward, since the feature vector entering the ML model is

built on averaging species-specific quantities. Though, the description of disordered and eutectic systems might require the incorporation of additional terms in the model, such as the entropy of mixing. We also expect that the ML approach can be improved further by including terms that can capture other sources of temperature-dependent contributions to the free-energy, e.g., configurational, electronic, or magnetic entropy.

In conclusion, we demonstrated that zero-Kelvin first-principles calculations can be augmented with ML models trained on experimental free energy slopes to facilitate the accurate yet computationally efficient prediction of high-temperature materials properties. As one example with relevance for chemical industry, we applied this concept to the pyrometallurgical reduction of 38 binary and ternary metal oxides, showing that ML-augmented first-principles calculations can predict oxide reduction temperatures with a mean absolute error of 64 K and correctly describe the temperature dependence of the reaction free energies. We further highlighted the importance of encoding and targeting physical properties that are directly related to the fundamental equations of thermodynamics and serve as a sensible prior to build accurate ML models. The approach is not limited to oxide and could also be applied to other classes of compounds, such as sulfides or nitrides, if at least some experimental reference data is available, since all model features are derived either from first principles (formation energies and bulk moduli), from the crystal structure, or from the periodic table. The proposed ML methodology can serve as a blueprint for the modeling of temperature-dependent materials processes with a manageable computational cost in cases where limited experimental data is available and may ultimately guide the design of novel materials and processes.

## Methods

**Details of the machine learning models.** All ML models were based on Gaussian Process Regression (GPR)[21] as implemented in CatLearn[32]. We built a Gaussian process (GP)

$$f(X) \sim \mathrm{GP}(\mathbf{0}, K(X, X')), \quad (5)$$

where $X$ defines the set of inputs $(x_i)$, $f(X)$ denotes the latent functions, and $K(X, X)$ is an $n \times n$ matrix with components $k(x_i, x_j)$ for a number of $n$ samples in the training or test set and $d$ dimensions of the input space.

The *kernel trick* is used to translate the input space into feature space with the covariance function $k(x, x')$. The kernel is applied to determine relationships between the descriptor vectors for candidates $x$ and $x'$. We used the radial basis function (RBF) kernel with the form

$$k(x, x') = \sigma_c^2 + \sigma_f^2 \exp\left(-\frac{\|x - x'\|^2}{2\ell^2}\right), \quad (6)$$

where $\sigma_f^2$ is a scaling factor, $\ell^2$ defines the kernel length scale, and $\sigma_c^2$ is a constant shift. The hyperparameters $\sigma_f^2$ and $\ell^2$ were optimized in each feature dimension of the fingerprint vector allowing for an anisotropic form of the kernel. This adds automatic relevance determination (ARD) capability to our model.

In the following, we refer to $X$ and $X_*$ as the accumulation of the feature vectors of the candidates in the training and test sets, respectively. The conditional distribution of the GP is given by the mean

$$\overline{f}(X_*) = K(X_*, X)\, K_y^{-1}\, \mathbf{y} \quad (7)$$

and covariance

$$\mathrm{cov}(\overline{f}(X_*)) = K(X_*, X_*) - K(X_*, X)\, K_y^{-1}\, K(X, X_*) \quad (8)$$

where $K_y = K(X, X) + \epsilon^2\, I$ is the $n \times n$ covariance matrix for the noisy target values $\mathbf{y}$ and noise level $\epsilon$. The variance for a new data $(\mathbf{x}_*)$ obtained from the training data $(X)$, is used to quantify the uncertainty of the process, is given by

$$\sigma^2(\mathbf{x}_*) = \mathbf{x}\lambda + K(\mathbf{x}_*, \mathbf{x}_*) - \mathbf{k}(\mathbf{x}_*)^T\, K_y^{-1}\, \mathbf{k}(\mathbf{x}_*), \quad (9)$$

where the $n \times 1$ covariance vectors between new data points and the training data $\mathbf{x}_i \in X$ are given by $\mathbf{k}(\mathbf{x}_*) = [K(\mathbf{x}_*, \mathbf{x}_1), \dots, K(\mathbf{x}_*, \mathbf{x}_n)]^T$. The predicted uncertainty is then given by $\sigma(\mathbf{x}_*)$. The first term applies the predicted noise to the uncertainty with $\mathbf{x}\lambda$ being the optimized regularization strength for the training data. We chose as initial hyperparameters $[\sigma_c, \sigma_f, \ell] = (1.0, 1.0, 1.0)$, with bounds on the noise level $\epsilon \in [1 \times 10^{-3}, 1 \times 10^{-1}]$ and performed an optimization of the hyperparameters

through maximizing the log marginal likelihood

$$\log p(\mathbf{y}|X, \boldsymbol{\theta}) = -\frac{1}{2}\mathbf{y}^T K_y^{-1}\mathbf{y} - \frac{1}{2}\log|K| - \frac{n}{2}\log 2\pi, \quad (10)$$

where $\boldsymbol{\theta}$ denotes the whole set of hyperparameters $(\sigma_c, \sigma_f, \ell,$ and $\epsilon)$. The hyperparameter optimization was performed using the Truncated Newton Constrained (TNC) method[33] as implemented in SciPy[34].

**Model validation.** The predictive power of our ML models was validated using three different techniques to ensure that the models are transferable to unseen data and do not exhibit overfitting. All reported error estimates were obtained from leave-one-out cross-validation (LOOCV), which is a standard method for assessing the generalization ability of models. This means, for each of the 38 oxides a GPR model was trained on a data set containing only data from the other 37 oxides. The error estimate was then determined by evaluating the model for the oxide that was left out. Thus, LOOCV establishes the transferability of the models for unseen data. Second, $k$-fold cross-validation was employed to detect correlations of the reference samples with each other by using test sets with increasing size. The results from the analysis confirm the error estimates from LOOCV and do not indicate any issues with the data set. See Supplementary Figs. 1 and 3 for further details. Finally, we varied the noise parameter of the GPR models over four orders of magnitude to determine whether the models were overfitted and reproduced unwanted information (such as noise) from the reference data or if they were able to robustly fit statical noisy observations from the reference data. As discussed in more detail in the Supplementary Methods section and shown in Supplementary Figs. 5 and 6, the goodness-of-fit of the models does not change significantly with the magnitude of the noise parameter, further indicating that the models do not suffer from over-fitting. Based on these three tests, the ML models are robust and exhibit good transferability.

**Model selection.** In addition to GPR, we confirmed that other regression models can also be successfully trained on the reference data. The accuracies (from LOOCV) obtained with a number of different regression models are discussed in the Supplementary Methods section and errors are compiled in Supplementary Table 5. As seen in the table, the models achieve overall similar accuracies as the GPR model, although the maximal errors are significantly larger, except for the three linear models (linear, ridge, and LASSO regression). Based on this test, we decided to proceed with Gaussian process regression, since it provides an uncertainty estimate for its predictions (as shown as error bars in Fig. 2).

**Density functional theory (DFT) calculations.** DFT calculations were performed using the Vienna Ab initio Simulation Package (VASP)[35,36] and the Strongly Constrained and Appropriately Normed (SCAN) exchange–correlation functional[37] including the dispersion corrections scheme rVV10[38]. The energy cutoff was 600 eV, and the Brillouin zone was sampled using the automatic $k$-mesh generation method implemented in VASP with 30 subdivisions in each direction. We used the pseudopotentials Li($s^1 p^0$), Na($p^6 s^1$), Mg ($s^2 p^0$), Al($s^2 p^1$), Si($s^2 p^2$), K(3s 3p 4s), Ca (3s 3p 4s), Ti ($d^3 s^1$), V ($p^6 d^4 s^1$), Cr ($d^5 s^1$), Mn ($d^6 s^1$), Fe (3p $d^7$ $s^1$), Co ($d^8 s^1$), Ni ($d^9 s^1$), Cu ($d^{10} p^1$), and Zn ($d^{10} p^2$) included in VASP version 5.4.

The convergence criterion for the electronic self-consistent cycle was fixed at $10^{-5}$ eV. Bulk structure optimizations were restarted until the optimizations required at most five ionic steps to ensure basis-set consistency with changing cell size and shape. For the $d$ bands of V, Cr, Mn, Co, and Ni in their respective oxides, a Hubbard-$U$[39,40] correction was used to counteract the self-interaction error and correct the oxide formation energies. The $U$ values were fitted to the experimental formation energy values[41] from the NIST-JANAF[28] and Cambridge DoITPoMs[29,30] thermodynamic tables as described in the Supplementary Methods section and visualized in Supplementary Fig. 7. We applied the following $U$ values: V (0.7 eV), Cr (1.5 eV), Mn (0.5 eV), Co (0.3 eV), and Ni (1.5 eV).

Phonon corrections were calculated using the *Phonons* module of the Atomic Simulation Environment (ASE)[42] package. We used $2 \times 2 \times 2$ supercells for calculating vibrational normal modes using the finite-displacement method[22].

**Entropy contributions.** The discussion of entropy effects above is limited to vibrational entropy, which is intuitively the largest entropy contribution for the considered oxide compounds at high temperature. For solid oxide solutions, the configurational entropy contributions to the free energy, $-T\Delta S_{\mathrm{mix}}$, can also become significant and are the origin of the stability of high-entropy oxides[43,44] and may determine the ground-state phase[45]. The entropy of mixing of an ideal solution is $\Delta S_{\mathrm{mix}} = -k_B \sum_{i=1}^{N} x_i \ln x_i$, where $k_B$ is Boltzmann's constant, $x_i$ is the concentration of species $i$, and $N$ is the number of species that share the same sublattice of the crystal structure. If it is known that a given oxide composition forms a solid solution, the contribution from the mixing entropy can be included analytically in the free energy of formation either on the level of first-principles theory or on top of the ML models. In case it is not known whether a composition is disordered or ordered, the tendency to disorder can be estimated based on first-principles calculations[46] of special quasirandom structures[47].

## Data availability

All experimental reference data is contained in Supplementary Table 1. Data from first-principles (SCAN) calculations that were generated in this study and enter the compound fingerprints are provided as Supplementary Software 1. Reference trained models have been deposited on GitHub and are publicly available from https://github.com/atomisticnet/gibbsml[49].

## Code availability

The ML methodology described in the present work was implemented in a Python package named GibbsML that provides the vectorized compound fingerprints and can be used to build regression models based on GPR as implemented in CatLearn[32]. GibbsML further makes use of the Atomistic Simulation Environment (ASE)[42] and the Python Materials Genomics (pymatgen) package[48]. The source code generated in this study has been deposited in GitHub under https://github.com/atomisticnet/gibbsml[49] and is publicly available under the terms of the MIT license. Additional source code that implements the model validation is provided as Supplementary Software 1. A web application for predicting free-energy curves Ellingham diagrams is hosted at: http://ellingham.energy-materials.org. Note that the web application interfaces with the Materials Project database[50] to extract first-principles data for the compound fingerprints, so that predictions are not limited to the scope of our own (SCAN-based) reference database.

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

## Acknowledgements

This work was supported by the National Science Foundation under Grant No. DMR-1940290 (Harnessing the Data Revolution, HDR). We acknowledge computing resources

from Columbia University's Shared Research Computing Facility project, which is supported by NIH Research Facility Improvement Grant 1G20RR030893-01, and associated funds from the New York State Empire State Development, Division of Science Technology and Innovation (NYSTAR) Contract C090171, both awarded April 15, 2010. The authors thank Mark S. Hybertsen, Dallas R. Trinkle, and Snigdhansu Chatterjee for helpful discussions.

## Author contributions

A.U. conceived and planned the project and supervised all aspects of the research. J.A.G.T., V.G., and A.U. developed the concept of the machine-learning methodology. J.A.G.T. implemented the method, compiled the reference data, performed the DFT calculations, constructed and validated the models, and analyzed the resulting data. The Hubbard-U parameters were determined by N.A. and J.A.G.T. N.A. also contributed to the conceptual development of the reaction free-energy formalism and the discussion of entropy contributions. T.H.E. developed the web application together with J.A.G.T. The manuscript was written by J.A.G.T., N.A., and A.U. and was read and revised by all authors.

## Competing interests

The authors declare no competing interests.
