## [Peer Review File · Nature Communications]

Augmenting zero-Kelvin quantum mechanics with machine learning for the prediction of chemical reactions at high temperaturesREVIEWER COMMENTS

Reviewer #1 (Remarks to the Author):

While the present study has used an emerging data analytics approach to predict high-temperature properties of metal oxides, a well-established computational method widely known as CALPHAD (CALculation of PHAse Diagram) has been successfully used in high-temperature metallurgy, such as the smelting process. Please see the references below:

Bale, C. W., et al. "FactSage thermochemical software and databases—recent developments." *Calphad* 33.2 (2009): 295-311.

Jung, In-Ho, and Marie-Aline Van Ende. "Computational Thermodynamic Calculations: FactSage from CALPHAD Thermodynamic Database to Virtual Process Simulation." *Metallurgical and Materials Transactions B* 51.5 (2020): 1851-1874.

Also, the present study shows that fairly good ML training could be obtained with a tiny dataset and very few feature sets. It very concerns that the model might have been overfitted. Perhaps it would be a good idea to make predictions for oxides that are not part of the training set to test the applicability of the present study. For example, it would be an interesting test that this approach can replicate the reduction temperature of oxide from CALPHAD with DFT formation energy and bulk modulus.

Reviewer #2 (Remarks to the Author):

I appreciate the chance to assess the paper.

The authors are doing good work. The area is new, and not many researchers to date have a depth of work behind their studies. This paper is an exception, with plenty under the hood.

Surprisingly (even for me) I actually don't really have technical comments. The actual 'work' part of the paper is sound, the methods are readily available, and the DFT is not controversial. However, I have a number of non-technical comments, which the authors may wish to consider in light of whether they edit anything technical.

Comments:

1) I am afraid to say that I was bamboozled by the paper title. The title is not fitting. One suggestion is a title without the word 'metallurgy', because this work is not really metallurgy as the readership may perceive at face value. I believe the use of that word will invoke readers that will be expecting metal-metal bonds, and, metallurgy in the context of structural materials. This is not that. The synch between the title and the paper could be improved. I suggest a more suitable title be found.

2) In the same vein as Comment 1, the paper as written hops contexts. The readers will not need convincing of various global shifts / impact statements. I think this paper would be significantly better, if the title was changed, and any other frills that detract from a neat story. The story is temp effects, and, applied to an important problem. The rest can go, and the paper re-ordered to be the strong punch the field needs. For example, I think this work will be much more broadly appreciated by those working in oxidation/corrosion (where there are thousands of researchers) as opposed to those recycling metals or pyro folks. I personally cant wait to use this for oxidation studies!

3) Minor comments:

-- remove in line 29 "often does not account for temperature effects", as there are some papers that do (even though not in the context of oxidation) - so best avoid

-- There is good scope for trimming ~20% of the intro, maybe more. I would jump straight into the relevant work, as there are now many papers that give the background context to why ML is important (even in this journal).

Finally, I want to make it clear I enjoyed the work, and my comments are not intended as un-movable criticism. I am sensing this is is also step 1, to future important work. I look forward to the journey.

Reviewer #3 (Remarks to the Author):

In this study, the authors tried to establish a general-purposed method to build accurate machine learning models for predicting the reduction temperatures of oxides and other compounds. The 0 K DFT calculation results are augmented with ML models trained on experimental free energy slopes. The idea of coupling features that can correctly capture the underlying thermodynamic principles is good. However, the present work is not ready for publication because of the following reasons:

(1) A dataset with 38 oxides reduction temperatures is too small, especially when authors used a relatively large number of "fingerprints" features in this dataset to train machine learning models. A machine learning model may be overfitted when trained with a small dataset having too many features.

(2) The authors tried to prove the fidelity of the machine learning by performing multiple rounds of cross-validation using different partitions to study the robustness of the predictive power with respect to the train/test fold size. However, considering the small data volume of the dataset, it is still very likely that the obtained models cannot well predict the data that is not included in this dataset.

(3) If one can get the Gibbs energy plot of oxides from Ellingham diagrams, why we don't determine the reduction temperature directly from the Ellingham diagram?

(4) Actually, the predicted free-energy curves from machine learning are more or less are constructed based on the DFT 0 K energy and the free-energy slopes from the Ellingham diagram. Then why we need machine learning to achieve this? why not directly constructing the "new" Ellingham diagram from DFT and the old diagram?

(5) Why only Gaussian process regression was considered in this work? Can other machine learning models also work?

Response to the Reviewers – Manuscript ID NCOMMS-21-19390

We thank both reviewers and the editor for their time spent on evaluating our manuscript. Detailed responses to all of the reviewers' questions are below. Additions to the revised manuscript are highlighted in blue, and removals are indicated in red.

Reviewer #1

While the present study has used an emerging data analytics approach to predict high-temperature properties of metal oxides, a well-established computational method widely known as CALPHAD (CALculation of PHase Diagram) has been successfully used in high-temperature metallurgy, such as the smelting process. Please see the references below:

*Bale, C. W., et al. "FactSage thermochemical software and databases—recent developments." *Calphad* 33.2 (2009): 295-311.*

*Jung, In-Ho, and Marie-Aline Van Ende. "Computational Thermodynamic Calculations: FactSage from CALPHAD Thermodynamic Database to Virtual Process Simulation." *Metallurgical and Materials Transactions B* 51.5 (2020): 1851-1874.*

Response: We thank the reviewer for the overall positive assessment of our manuscript. We added the two references suggested by the reviewer and a remark on virtual process design with CALPHAD to the paragraph that discusses the CALPHAD method (page 3, paragraph 2).

CALPHAD is the method of choice for the construction of phase diagrams when assessed thermodynamic information for all relevant phases is available. Our work demonstrates that the temperature dependence of the free energy of reaction can be cross-learned, so that predictions for unseen oxides can be made even when no phase diagram data is available. The approach is thus fully complementary to CALPHAD.

Also, the present study shows that fairly good ML training could be obtained with a tiny dataset and very few feature sets. It very concerns that the model might have been overfitted. Perhaps it would be a good idea to make predictions for oxides that are not part of the training set to test the applicability of the present study. For example, it would be an interesting test that this approach can replicate the reduction temperature of oxide from CALPHAD with DFT formation energy and bulk modulus.

Response: We agree with the reviewer that it is important to test for overfitting of ML models by establishing the transferability of the models to unseen data, and testing for overfitting was a priority for us when we prepared the original manuscript. The error quantification reported in the manuscript and the predictions shown in Figures 2, 3, and 4 are therefore based on leave-one-out cross-validation (LOOCV). This means, 38 ML models were trained each using data from

37 of the oxides while withholding data from the 38th oxide. Only predictions for the oxides not included in the training were analyzed, and thus only true predictions were reported.

We recognize that it was not explicitly stated in the original manuscript that all validation was done for unseen data, and we have extended the statement mentioning LOOCV in the revised manuscript (page 9, second paragraph) to read

[We trained a GPR-based ML model on the experimental reduction temperatures of Table S1 and quantified its accuracy using leave-one-out cross-validation (LOOCV).] LOOCV ensures that the model is evaluated only for samples that were not used for training and is a standard technique for assessing the transferability of a model (see further details in the method section).

To further ensure that the LOOCV estimate was not biased by a strong correlation of some of the samples in our database, we also performed k -fold cross-validation (CV) with different test/train

Figure R1 Model accuracy for predicting reduction temperatures when varying the GP noise parameter. Comparison between experimental and predicted formation free-energy changes when building the GP models using the following parameter values: (a) 10^{-1} , (b) 10^{-2} , (c) 10^{-3} and (d) 10^{-4} .

splits (Figure S3 in the supporting information). If the models were overfitted, the error in k-fold CV would rapidly increase with increasing test/train ratio, which is not seen for our data. Note that some increase of the error is expected due to the decreasing train set size.

In response to the reviewer's question, we performed additional tests specifically for overfitting. Overfitting occurs when a regression model represents also the noise in the reference data instead of only the ground truth. To minimize the likelihood of overfitting, we included a homoscedastic noise term to the kernel of the Gaussian process regression (GPR) models (the noise level is denoted ϵ in the methods section). If the models were indeed overfitted, their goodness of fit would vary strongly with the value of the noise parameter ϵ . As seen in **Figure R1**, even when the noise parameter is varied over four orders of magnitude, the goodness of fit does not significantly change. A more detailed discussion of this analysis has been added as new section S7 to the supporting information.

To summarize all of the above validation steps concisely for the reader, we added a brief new subsection to the methods section (page 19 of the revised manuscript):

Model validation

The predictive power of our ML models was validated using three different techniques to ensure that the models are transferable to unseen data and do not exhibit overfitting. All reported error estimates were obtained from leave-one-out cross-validation (LOOCV), which is a standard method for assessing the generalization ability of models. This means, for each of the 38 oxides a GPR model was trained on a data set containing only data from the other 37 oxides. The error estimate was then determined by evaluating the model for the oxide that was left out. Thus, LOOCV establishes the transferability of the models for unseen data. Second, k-fold cross-validation was employed to detect correlations of the reference samples with each other by using test sets with increasing size. The results from the analysis confirm the error estimates from LOOCV and do not indicate any issues with the data set. See Section S2 in the supporting information for further details. Finally, we varied the noise parameter of the GPR models over four orders of magnitude to determine whether the models were overfitted and reproduced unwanted information (such as noise) from the reference data or if they were able to robustly fit statical noisy observations from the reference data. As discussed in detail in Section S7 of the supporting information, the goodness-of-fit of the models does not change significantly with the magnitude of the noise parameter, further indicating that the models do not suffer from overfitting. Based on these three tests, the ML models are robust and exhibit good transferability.

We thank the reviewer again for motivating us to discuss the model validation in greater detail. We believe that the additions made in the revised manuscript and in the supporting information conclusively addressed the reviewer's concerns about overfitting.

Reviewer #2

I appreciate the chance to assess the paper.

The authors are doing good work. The area is new, and not many researchers to date have a depth of work behind their studies. This paper is an exception, with plenty under the hood.

Surprisingly (even for me) I actually don't really have technical comments. The actual 'work' part of the paper is sound, the methods are readily available, and the DFT is not controversial. However, I have a number of non-technical comments, which the authors may wish to consider in light of whether they edit anything technical.

Response: We thank the reviewer for the very positive evaluation of our manuscript and for the many helpful comments that he offered. We revised the manuscript following the reviewer's suggestions.

Comments:

1) I am afraid to say that I was bamboozled by the paper title. The title is not fitting. One suggestion is a title without the word 'metallurgy', because this work is not really metallurgy as the readership may perceive at face value. I believe the use of that word will invoke readers that will be expecting metal-metal bonds, and, metallurgy in the context of structural materials. This is not that. The synch between the title and the paper could be improved. I suggest a more suitable title be found.

Response: The reviewer makes an important point. Although we maintain that pyrometallurgy is an aspect of metallurgy, the core message of our manuscript is indeed more general. In response, we changed the title of the revised manuscript to:

Augmenting zero-Kelvin quantum mechanics with machine learning for the prediction of chemical reactions at high temperatures

2) In the same vein as Comment 1, the paper as written hops contexts. The readers will not need convincing of various global shifts / impact statements. I think this paper would be significantly better, if the title was changed, and any other frills that detract from a neat story. The story is temp effects, and, applied to an important problem. The rest can go, and the paper re-ordered to be the strong punch the field needs. For example, I think this work will be much more broadly appreciated by those working in oxidation/corrosion (where there are thousands of researchers) as opposed to those recycling metals or pyro folks. I personally cant wait to use this for oxidation studies!

Response: We thank the reviewer for pointing out the relationship of our work with oxidation/corrosion. We agree that the introduction of the paper could be made more general and more focused, and we have rewritten it in part in response to this question. Please see the introduction section (pages 2-4) of the marked-up version of the revision for highlighted changes.

3) *Minor comments:*

-- remove in line 29 "often does not account for temperature effects", as there are some papers that do (even though not in the context of oxidation) - so best avoid

-- There is good scope for trimming ~20% of the intro, maybe more. I would jump straight into the relevant work, as there are now many papers that give the background context to why ML is important (even in this journal).

Response: We followed the reviewer's suggestion and removed the statement regarding temperature effects and substantially trimmed the introduction. We agree that the importance of ML has been previously established and does not need to be reiterated. We thank the reviewer for this suggestion.

Finally, I want to make it clear I enjoyed the work, and my comments are not intended as unmovable criticism. I am sensing this is also step 1, to future important work. I look forward to the journey.

Response: we appreciate your supportive and constructive review of our work and believe that the revision in response to your questions/suggestions has further improved the quality of our manuscript.

Reviewer #3

In this study, the authors tried to establish a general-purposed method to build accurate machine learning models for predicting the reduction temperatures of oxides and other compounds. The 0 K DFT calculation results are augmented with ML models trained on experimental free energy slopes. The idea of coupling features that can correctly capture the underlying thermodynamic principles is good.

Response: We thank the reviewer for the encouraging review. We address all of the reviewer's concerns point by point in the following.

*However, the present work is not ready for publication because of the following reasons:
(1) A dataset with 38 oxides reduction temperatures is too small, especially when authors used a relatively large number of "fingerprints" features in this dataset to train machine learning models. A machine learning model may be overfitted when trained with a small dataset having too many features.*

Response: We agree with the reviewer that feature selection is an important aspect of model construction that could have been discussed more extensively in our original submission. To determine redundant features that do not contain additional information and could potentially

Figure R2 Variation of the Mean Absolute Error (MAE) when excluding features compared to the MAE when all the features were included (in percentage).

lead to overfitting, we evaluated the change of the mean absolute error (MAE) when individual features and combinations of features are removed from the feature vector (**Figure R2**). Note that this is a standard approach for feature selection sometimes referred to as *recursive feature elimination*. As seen in **Figure R2**, the MAE decreases by nearly 20% when the space group is excluded from the set of features, indicating that it did not contain any information that was not already represented by the other features and led to overfitting. The combination of excluding the space group and oxidation state is shown as one example of removing two features. Our feature selection process is detailed in the new Section S4 in the supporting information. We also added the following additional statement to page 9 of the revised manuscript:

[The construction of the compound fingerprint by combining the above properties is described in supplementary Section S3.] We employed recursive feature elimination to detect redundant features and avoid overfitting as is detailed in supplementary Section S4.

(2) The authors tried to prove the fidelity of the machine learning by performing multiple rounds of cross-validation using different partitions to study the robustness of the predictive power with respect to the train/test fold size. However, considering the small data volume of the dataset, it is still very likely that the obtained models cannot well predict the data that is not included in this dataset.

Response: The model validation reported in the original manuscript, *i.e.*, leave-one-out cross-validation (LOOCV) and *k*-fold cross-validation, are testing precisely the transferability of our model to unseen data. In the revision, we introduced a new subsection in the method section that explains the different cross-validation strategies and why they together provide robust error estimates on unseen data. Please see also our response to reviewer #1.

For the revision, we performed also an additional overfitting test by varying the regularization parameter of the Gaussian process models over four orders of magnitude. The results are shown in **Figure R1** and in a new Section S7 in the supporting information, and there is no evidence of overfitting. Please see the response to reviewer #1 for further details.

In conclusion, the reported model accuracies are indeed for unseen data, and based on the results from different tiers of validation there is no indication that the model would fail for other binary or ternary metal oxides.

(3) If one can get the Gibbs energy plot of oxides from Ellingham diagrams, why we don't determine the reduction temperature directly from the Ellingham diagram?

Response: The experimental reduction temperatures used in the present work were, in fact, determined from experimentally measured reaction free energies, *i.e.*, from Ellingham diagrams. When such data is available, there is no need for any additional modeling. The key message of our work is that it is possible to cross-learn reaction free energies, so that the energies of unseen oxides can be predicted using a model that was trained on other metal oxides. We believe that we have demonstrated conclusively that a transferable ML model of the free-energy slopes can

be trained, so that Ellingham diagrams (and thus reduction temperatures) can be predicted, with good accuracy, for unseen oxides and only using data from zero-Kelvin first principles calculations.

We further clarified the origin of our reference data in the introduction section of the revised manuscript (page 3):

For example, we were only able to compile a set of 38 metal oxide reduction temperatures from public data sources that were extracted from experimentally measured free energies of reaction (see Supplementary Table S1).

Additionally, we reworded the conclusion statement to make it clearer that our model can predict reduction temperatures of unseen oxides (page 17, line 309 and following)

As one example with relevance for chemical industry, we applied this concept to the pyrometallurgical reduction of 38 binary and ternary metal oxides, showing that ML-augmented first-principles calculations can predict ~~oxide~~ the reduction temperatures of unseen oxides with a mean absolute error of 64 K and correctly describe the temperature dependence of the reaction free energies.

(4) Actually, the predicted free-energy curves from machine learning are more or less are constructed based on the DFT 0 K energy and the free-energy slopes from the Ellingham diagram. Then why we need machine learning to achieve this? why not directly constructing the "new" Ellingham diagram from DFT and the old diagram?

Response: The reviewer's question is related to the previous point. Our ML/DFT hybrid model can predict Ellingham diagrams of oxides that were not included in the training set. As discussed above, the reported results are predictions on unseen data. The revised introduction section and the additional "model validation" section state this now explicitly to avoid any misunderstandings. This approach works robustly because DFT provides accurate 0-Kelvin enthalpies and variations in the temperature dependent slope are simple enough that 37 data points are sufficient. As so often with machine learning models, the accuracy can likely be further improved as additional reference data becomes available.

(5) Why only Gaussian process regression was considered in this work? Can other machine learning models also work?

Response: The reviewer raises an interesting question. We initially chose Gaussian process regression because it is well suited for moderately-sized data sets and provides out-of-the-box a fine-grained uncertainty estimate (error bars) for predictions, which is in our opinion extremely valuable. For the revision we have confirmed that other regression models can also be successfully trained on the reference data. The accuracies (from leave-one-out cross-validation) obtained with a number of different regression models are given in **Table R1** (new supporting Table S5).

Table R1 Error estimates for different regression models. The mean absolute error (MAE) and the root mean squared error (RMSE) were determined by leave-one-out cross-validation. The final row shows the maximum error (in reduction temperature).

Regression Model	RMSE (K)	MAE (K)	Max. Error (K)
Linear	75	61	176
Ridge	77	62	191
LASSO	78	61	181
ElasticNet	80	63	223
Support Vector	147	127	376
Kernel Ridge	69	57	165
Random Forest	93	69	289
Gradient Boosting	74	56	250
AdaBoost	91	58	299
ExtraTrees	75	59	243

As seen in the table, the models achieve overall similar accuracies as the Gaussian process regression model, although the maximal errors are significantly larger, except for the three linear models (linear, ridge, and LASSO regression). Only support vector regression shows an unexpectedly large error. Based on this test, the Gaussian process model is still preferable, since it provides an uncertainty estimate for its predictions.

In the revised manuscript, we included a brief new subsection on model selection in the methods section containing the above discussion:

Model selection

In addition to GPR, we confirmed that other regression models can also be successfully trained on the reference data. The accuracies (from LOOCV) obtained with a number of different regression models are given in supporting Table S5. As seen in the table, the models achieve overall similar accuracies as the GPR model, although the maximal errors are significantly larger, except for the three linear models (linear, ridge, and LASSO regression). Based on this test, we decided to proceed with Gaussian process regression, since it provides an uncertainty estimate for its predictions (as shown as error bars in Figure 2).

We thank the reviewer again for their helpful comments and suggestions. Including the additional discussion of feature and model selection has, in our opinion, further improved our manuscript.

REVIEWERS' COMMENTS

Reviewer #2 (Remarks to the Author):

The revised paper has addressed the reviewer comments, and has also gone further in improving the overall readability with regards to holistic modifications from all three reviewers. I recommend acceptance of the work.

I also reviewed the revised paper with regards to whether Reviewer 3 concerns were addressed. Both in the response/rebuttal, and in the edited paper, I believe the comments of Reviewer 3 were adequately addressed. The key comments regarding small dataset, and what that means, have been well addressed. There is no 'workaround' for a small dataset - and that would not be a basis for rejection (not that I am suggesting that was proposed). In fact, this work, with a small dataset (which if compared to other materials studies, is actually a very BIG dataset, so its all relative) is instructive to the field with regards to how to handle such a scenario. The authors present methods that are meaningful, and to be honest - I will adopt in my own research (noting I do not know the authors and only aware of the work through this review process, so I am in no way favourably disposed, just simply being honest).